# The Effect of the Barrier Layer on the Uniformity of the Transport Characteristics of AlGaN/GaN Heterostructures on HR-Si(111)

**DOI:** 10.3390/mi15040536

**Published:** 2024-04-16

**Authors:** Yujie Yan, Yangbowen Liu, Guodong Xiong, Jun Huang, Bing Yang

**Affiliations:** Hubei Jiufengshan Laboratory, Wuhan 430074, China; yanyujie@jfslab.com.cn (Y.Y.); liuyangbowen@jfslab.com.cn (Y.L.); xiongguodong@jfslab.com.cn (G.X.); yangbing@jfslab.com.cn (B.Y.)

**Keywords:** AlGaN/GaN heterostructures, high-resistivity silicon, transport characteristics, uniformity, polarization effect

## Abstract

The high transport characteristics of AlGaN/GaN heterostructures are critical components for high-performance electronic and radio-frequency (RF) devices. We report the transport characteristics of AlGaN/GaN heterostructures grown on a high-resistivity (HR) Si(111) substrate, which are unevenly distributed in the central and edge regions of the wafer. The relationship between the composition, stress, and polarization effects was discussed, and the main factors affecting the concentration and mobility of two-dimensional electron gas (2DEG) were clarified. We further demonstrated that the mechanism of changes in polarization intensity and scattering originates from the uneven distribution of Al composition and stress in the AlGaN barrier layer during the growth process. Furthermore, our results provide an important guide on the significance of accomplishing 6 inch AlGaN/GaN HEMT with excellent properties for RF applications.

## 1. Introduction

AlGaN/GaN high-electron-mobility transistors (HEMTs) are widely applied to power and RF electrics due to their high electron density, high electron mobility, and excellent power and frequency characteristics [1,2,3,4,5,6,7]. Attributed to the strong polarization effect at the interface between AlGaN and GaN, high 2DEG density forms without additional doping. The difference in lattice constant in crystal causes accumulation of polarized charges and forms deep electron potential wells at the heterointerface [8,9]. Moreover, the complicated technological requirements and prohibitive costs of gallium nitride (GaN) single-crystal substrates greatly limit the development of homoepitaxy, which also means heteroepitaxy has become the mainstream process for AlGaN/GaN HEMTs [10,11,12]. Among these devices, silicon substrates are highly favored due to their low cost and compatibility with CMOS processes, while they also make it possible to grow large-size GaN films and greatly reduce the cost of devices [13,14,15].

One of the key focal directions for high-power and high-frequency electronic devices is optimizing the transport characteristics of AlGaN/GaN heterostructures. However, due to the significant lattice mismatch and thermal mismatch between the GaN and the Si substrate, it is difficult to obtain high-quality and crack-free GaN epitaxial films as the Si substrate size and the epi-layer thickness increase [16]. During the growth process, they are prone to various problems appearing inside each layer, including defects, residual stress, and wafer bow [17,18]. Especially in the RF field, the high-resistivity Si (HR-Si) substrate is thought to substitute the SiC for its low RF loss and cost as advantages. In comparison to low-resistance Si substrates, high-resistance Si substrates are not considered optimal for the growth of high-quality AlGaN/GaN heterojunctions due to their inferior mechanical properties. Hence, an in-depth analysis of the uniformity of electrical properties of AlGaN/GaN heterostructures on HR-Si is indispensable to enhance device performance, reduce the cost of radio frequency devices, and improve their yields. A common method used for alleviating stress and dislocations is to optimize buffer structures, such as AlN [19,20], AlGaN/GaN superlattices [21,22], and compositionally graded AlGaN layers [23,24,25]. Moreover, the uniformity of the transport characteristics of AlGaN/GaN heterostructures also cannot be ignored. Recent research has shown that the electrical properties of heterostructures are affected by wafer bow [26] and different buffer layer structures [27,28]. To illustrate the effect of the uniformity of sheet resistance, J. Ma et al. studied the relationship between the stress, dislocations, and transport characteristics of AlGaN/GaN heterostructures, and also attributed the nonuniformity to the uneven distribution of stress [29]. Nonetheless, the mechanism acting on the homogeneity of transport properties has been poorly studied, and requires in-depth understanding. In this work, we investigate the uniformity of 2DEG density and electron mobility at different wafer positions by analyzing the relationship between crystal structure, strain, and the polarization effect, which is crucial for further improving the transport characteristics of AlGaN/GaN HEMTs on the Si substrate and provides a significant guide for improving epitaxial film quality.

In this paper, the transport properties of AlGaN/GaN HEMT on the HR-Si substrate and the mechanism of the diversification of 2DEG density and electron mobility are investigated. Structural and optical characterization techniques were used to quantify crystal quality and stress, and then the polarization charge density at the heterointerface was calculated based on quantization data. Our results show that polarization effects strongly interrelate to the Al content, strain in the AlGaN, and residual stress in GaN, which affects the distribution of 2DEG density and electron mobility. Furthermore, electron mobility is greatly influenced by scattering, the Al content, and polarization charges at the AlGaN/GaN heterointerface, which also affect the distribution of electron mobility.

## 2. Experimental Details

The AlGaN/GaN HEMT structure was grown on a 6-inch high-resistance Si substrate (HR-Si(111)) with metal organic chemical vapor deposition (MOCVD, Aixtron AIX G5+, AIXTRON, Herzogenrath, Germany). The schematic structure is shown in Figure 1a, and the layer structure of the cross-section was further observed specifically by scanning transmission electron microscopy (STEM). In order to characterize the uniformity of the structure and the transport properties, nine measurement locations were selected in the center and edge regions of the 6-inch wafer, as shown in Figure 1b. The transport properties were analyzed using a non-contact Hall measurement (N-Hall). The crystalline quality, chemical composition, and stress were characterized using high-resolution X-ray diffraction (HR-XRD). A wafer bow test was performed by using the Flatness Analyzer measurement. Atomic force microscopy (AFM, with a tapping mode) and STEM were performed to investigate the surface morphology, microstructure, and dislocation distribution.

## 3. Results and Discussion

In order to observe the transport properties and distribution of AlGaN/GaN heterostructures, we conducted electrical measurements at nine locations (see Figure 1b), including sheet resistance (*R_s_*), electron mobility (*μ_n_*), and 2DEG density (*n_s_*), as shown in Figure 2. The transport characteristics of the entire epitaxial film are unevenly distributed. Compared to the edge region of the epitaxial film, the electron mobility in the central region is higher, while the distribution trend of 2DEG density is the opposite. A strong correlation between 2DEG density and electron mobility at the heterointerface has been reported [27,30], which is in agreement with our work. Here, we mainly discuss the uniformity of electron density and mobility. 

Figure 3(a1–a3) shows the surface morphology and contours of the epitaxial film in the center and edge regions to characterize the uniformity of surface roughness. The measurement locations were selected from the regions 1, 2, and 6 in Figure 1b, with 5 μm × 5 μm AFM scans. The results show that the surface is smooth and flat, without obvious defects such as pores and cracks. Root mean square roughness (RMS) is an important indicator for evaluating surface roughness and morphology. The average value of RMS is 0.23 nm, with clear atomic-level step flow, which also confirms that the AlGaN/GaN HEMT structure grows in two-dimensional (2D) layered mode. Furthermore, annular dark-field (ADF) imaging by scanning transmission electron microscopy (STEM) shows the details of dislocations distribution, under two-beam conditions with g = [0002] and g = [11$\bar{2}$0], respectively, as shown in Figure 3(b1,b2,c1,c2). Different g vectors were used to characterize dislocation types, where g = [0002] represents screw and mixed dislocations, and g = [11$\bar{2}$0] represents edge dislocations and mixed-type dislocations [31]. As a result, only a small number of threading dislocations continue extending, and most dislocations bend at the buffer interface or during the growth process. The interaction of dislocations forms dislocation loops, leading to dislocation annihilation, effectively reducing the dislocation density.

The crystalline quality and stress of the epitaxial film were evaluated by HR-XRD at nine positions (see Figure 4), suitable for investigating the influence of crystal quality on transport properties. Figure 4 shows ω scans of rocking curves (RCs) towards the symmetric (002) and asymmetric (102) planes, of which full-width at half-maximum (FWHM) can reflect the crystalline quality [32]. The values of FWHM at different positions are presented in Table 1, indicating the high uniformity of crystal quality in the center and edge regions, especially the (002) plane. It is worth noting that for the (102) plane, the FWHM in the central region is slightly higher, which is related to stress distribution.

Figure 5(a1,a2) shows the 2θ − ω scan along the (002) and (102) planes used to measure lattice constant *c* and *a*, respectively, and the diffraction peak of AlGaN is located between GaN and AlN. For hexagonal systems [33], the lattice constant (*a* and *c*) is expressed as: (1)$$\frac{1}{d^{2}}=\frac{4}{3}\frac{\left(h^{2}+hk+k^{2}\right)}{a^{2}}+{\left(\frac{l}{c}\right)}^{2}$$
where *d* represents the interplanar spacing of the probed lattice plane (*hkl*). According to Vegas’s law [34], as a ternary alloy of GaN and AlN, the lattice constant of Al*_x_*Ga_1−*x*_N depends on the Al content *x*, which can be expressed as:$$c_{AlGaN}=c_{AlN}\cdot x+c_{GaN}\cdot \left(1-x\right)$$
(2)$$a_{AlGaN}=a_{AlN}\cdot x+a_{GaN}\cdot (1-x)$$

As a result, the Al content *x* was calculated based on the lattice constant *a* or *c* of GaN/AlGaN/AlN in accordance with Equation (2), as shown in Figure 5(b1). It is clear to see that the uniformity of the Al content varies with different measurement locations. The Al content in the central region of AlGaN/GaN heterostructures (~0.27) is lower than that in the edge region (~0.3), which may be related to the strain state of the Al*_x_*Ga_1−*x*_N barrier layer and the migration rate of Al atoms during the growth process [35].

According to Bragg’s law defined as: 2dsin⁡θ=nλ, it is known that the shift of diffraction peaks indicates the change in the lattice constant. The positions of diffraction peaks of GaN and AlGaN both shift at different measurement regions (see Figure 5(a1,a2)), which demonstrates that lattice distortion may occur in the GaN and AlGaN layers due to lattice mismatch. In hexagonal crystal systems, strain can be defined as in-plane strain ($\varepsilon_{xx}$) and out-of-plane strain ($\varepsilon_{zz}$) [36], expressed as: $$\varepsilon_{xx}=\varepsilon_{yy}=\frac{a-a_{0}}{a_{0}}\;\;\;\varepsilon_{zz}=\frac{c-c_{0}}{c_{0}}$$
(3)$$2\frac{c_{13}}{c_{33}}\frac{a-a_{0}}{a_{0}}=-\frac{c-c_{0}}{c_{0}}$$

The in-plane stress $\sigma_{xx}\;$ is related to $\varepsilon_{xx}$ by Hooke’s law [37]: (4)$$\sigma_{xx}=\left[\left(C_{11}+C_{12}\right)-2\frac{C_{13}^{2}}{C_{33}}\right]\cdot \varepsilon_{xx}$$
where, $a_{0}\;{and\;c}_{0}$ are the lattice constant of strain-free bulk materials, $a_{GaN}$ = 3.189 Å, $c_{GaN}$ = 5.186 Å; $a_{AlN}$ = 3.112 Å, and $c_{AlN}$ = 4.978 Å [36]. *a* and *c* are experimental lattice parameters, *C_ij_* represents the elastic constants. The piezoelectric coupling matrices and components of the elastic constant matrix for the materials are listed in Table 2 [38]. Moreover, Al*_x_*Ga_1−*x*_N-related parameters are linear interpolation sets between the physical properties of GaN and AlN according to Vegas’s law. 

Figure 5(b2) displays the in-plane stress of AlGaN and GaN calculated using Equations (3) and (4). For the entire AlGaN/GaN HEMT, the residual stress of GaN is not uniformly distributed, the central region is under tensile stress, while the edge region is under tensile or compressive stress, which may be related to the uniformity of the material and the dislocation distribution [32]. This also affects the stress of the AlGaN barrier layer grown on top of the GaN. Due to the smaller lattice constant of AlGaN compared to GaN, a thin layer of AlGaN is grown pseudo-morphically on the GaN–substrate template, which results in the generation of tensile stress. Furthermore, there are variations in the magnitude of the tensile stress at different positions, with the central region having less tensile stress than the edge.

Figure 6 shows the wafer warp of the AlGaN/GaN HEMT on the HR-Si substrate, measured using Flatness Analyzer. The warp value is −27.574 μm. The inset image has enabled us to better represent the 3D morphology of the epitaxial wafer when in its warped state. The wafer has a concave downward curvature, which is akin to a bowl shape and is a manifestation of the tensile stress it endures. Larger warpage will cause differences in dislocation densities at different locations in the epitaxial layer, which affects the uniformity of sheet resistance [26].

The 2DEG at the AlGaN/GaN heterointerface strongly depends on spontaneous polarization (*P_SP_*) and piezoelectric polarization (*P_E_*), mainly derived from the structural characteristics and stress state of the material itself. From the previous discussion, it can be concluded that both AlGaN and GaN are affected by stress. The *P_E_* in strain layers can be obtained by Equations (3) and (5), as follows [38]:$$P_{E}=e_{33}\varepsilon_{zz}+e_{31}\left(\varepsilon_{xx}+\varepsilon_{yy}\right)$$
(5)$$P_{E}=2\frac{a-a_{0}}{a_{0}}\left(e_{31}-e_{33}\frac{c_{13}}{c_{33}}\right)$$
where *e_ij_* and *C_ij_* are represented as piezoelectric coefficients and elastic constants, respectively, as shown in Table 2. Due to $e_{31}-e_{33}\frac{c_{13}}{c_{33}}<0$, *P_E_* is negative for tensile stress and positive for compressive stress. Figure 7 shows the polarization effect of strain on AlGaN and GaN, respectively. It can be seen that *P_SP_* plays a lead role and increases with the increase in Al content, while *P_E_* increases with increasing tensile stress in AlGaN (see Figure 7a). However, the thicker GaN layer causes more stress to be relaxed, and as a result, the contribution of *P_E_* to the total polarization effect is smaller compared to *P_SP_*. A clearer illustration of the changes in the *P_E_* of strain in GaN is shown in Figure 7b. At different positions, *P_E_* exhibits negative or positive values due to tensile or compressive stress. 

Subsequently, Figure 8a shows the relationship between 2DEG density (blue dotted line) and the total polarization. For Ga-face AlGaN/GaN heterostructures, the fixed polarization charge density σ is represented as:(6)$$\sigma =P_{AlGaN}-P_{GaN}=\left\{P_{SP}\left(AlGaN\right)+P_{PE}\left(AlGaN\right)\right\}-\left\{P_{SP}\left(GaN\right)+P_{PE}\left(GaN\right)\right\}$$

The variation trend of 2DEG density is broadly in accordance with the total polarization, indicating that 2DEG density is jointly affected by the polarization effect in AlGaN and GaN, where the contribution of AlGaN is particularly significant. The uniformity of Al content and stress in the AlGaN barrier layer will directly affect the density and distribution of 2DEG. 

As shown in Figure 8b, electron mobility decreases with the increase in Al content at different positions. It is well known that electron mobility is strongly influenced by several kinds of scattering effects, including alloy disorder scattering, interface rough scattering, and polar optical phonon scattering. The increase in Al content will cause the roughness of the AlGaN/GaN heterointerface to increase, resulting in enhanced interface roughness scattering. Moreover, the increase in 2DEG density caused by increasing Al content also leads to changes in various scattering effects [39].

## 4. Conclusions

In conclusion, our work investigated the transport properties of an AlGaN/GaN heterostructure grown on an HR-Si substrate and proposed that the Al content and stress in the AlGaN barrier layer are the main factors affecting the uniformity of transport properties. By calculating the polarization charge density of the AlGaN/GaN heterointerface, it was confirmed that the uneven distribution of Al content and the stress in AlGaN, as well as the residual stress in GaN affect the density and distribution of 2DEG. Simultaneously, the Al content also affects the uniformity of electron mobility. Therefore, we have demonstrated the importance of the uniformity of the AlGaN barrier layer for the transport properties of the AlGaN/GaN heterostructure, which is meaningful for the development of over 200 mm GaN-on-Si technology.

## Figures and Tables

**Figure 1 micromachines-15-00536-f001:**
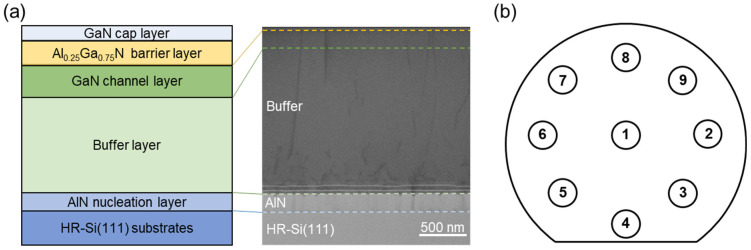
(**a**) Schematic illustration of AlGaN/GaN HEMT structure grown on Si(111) substrates, and STEM-BF image of cross-sectional structure. (**b**) Nine points measurement on 6-inch wafer.

**Figure 2 micromachines-15-00536-f002:**
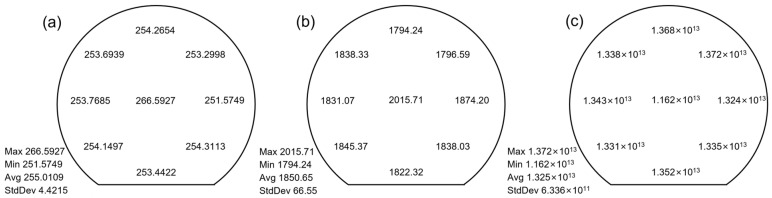
The transport properties of the 6-inch epitaxial film were examined at nine locations. (**a**) Sheet resistance, (**b**) electron mobility, and (**c**) 2DEG density.

**Figure 3 micromachines-15-00536-f003:**
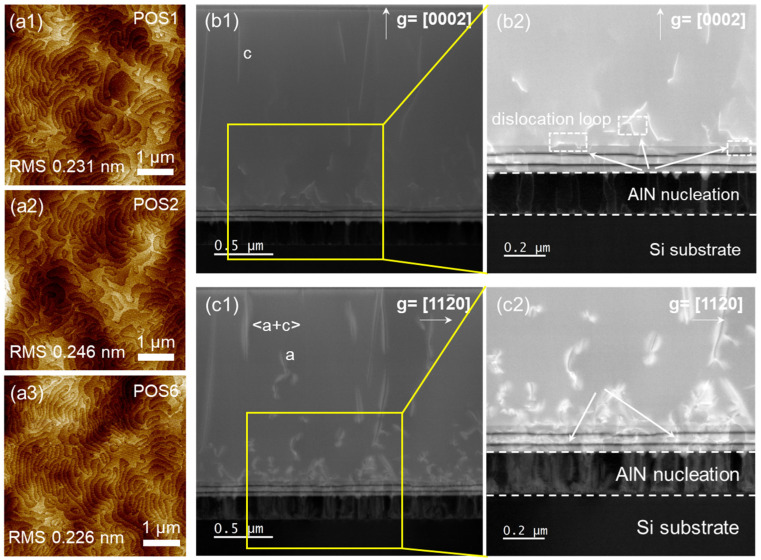
(**a1**–**a3**) The AFM images at different measurement locations. Cross-sectional STEM-ADF images of AlGaN/GaN HEMT structures grown on Si, obtained with diffraction vectors g = [0001] (**b1**,**b2**) and g = [11$\bar{2}$0] (**c1**,**c2**).

**Figure 4 micromachines-15-00536-f004:**
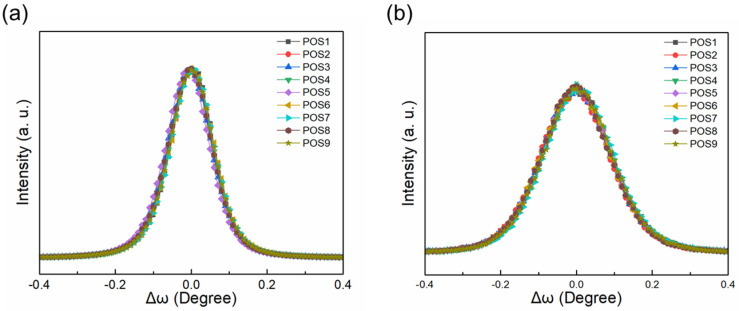
The ω scans of the rocking curves (RCs) scan towards the (002) plane (**a**) and the (102) plane (**b**).

**Figure 5 micromachines-15-00536-f005:**
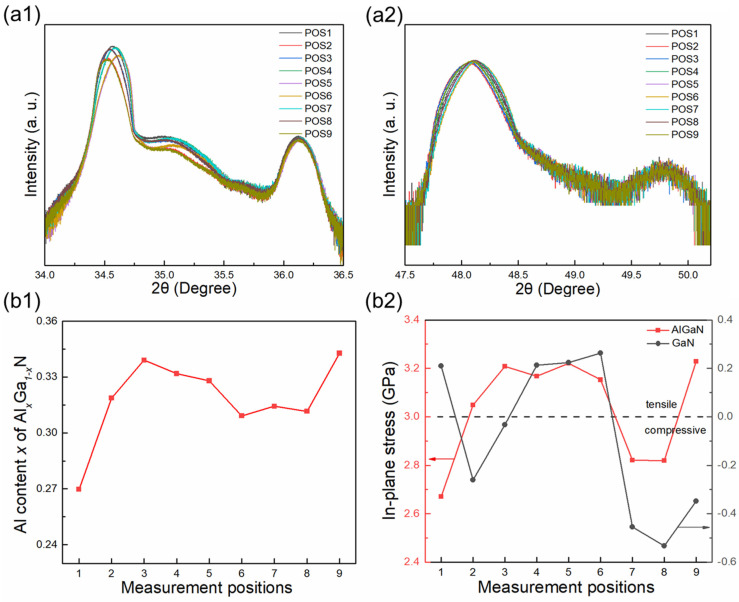
The 2θ − ω scan along the (002) (**a1**) and (102) (**a2**) planes. (**b1**) The Al content *x* of the Al*_x_*Ga_1−*x*_N barrier layer at different measurement locations. (**b2**) The in-plane stress of the GaN layer and the Al*_x_*Ga_1−*x*_N barrier layer.

**Figure 6 micromachines-15-00536-f006:**
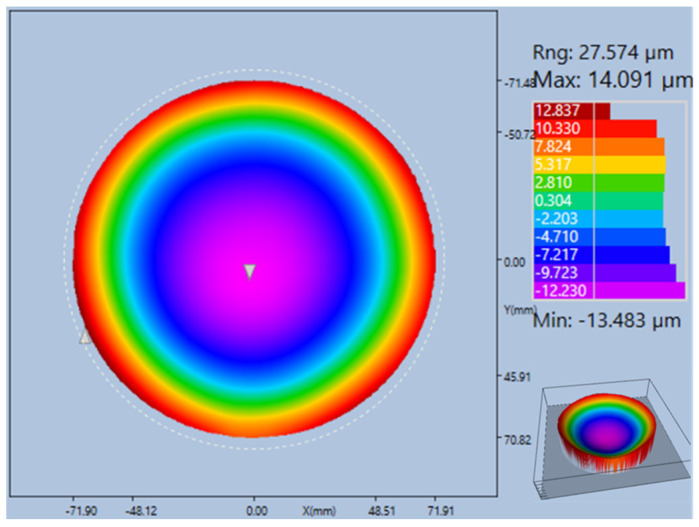
Wafer warp of AlGaN/GaN HEMT grown on the Si substrate.

**Figure 7 micromachines-15-00536-f007:**
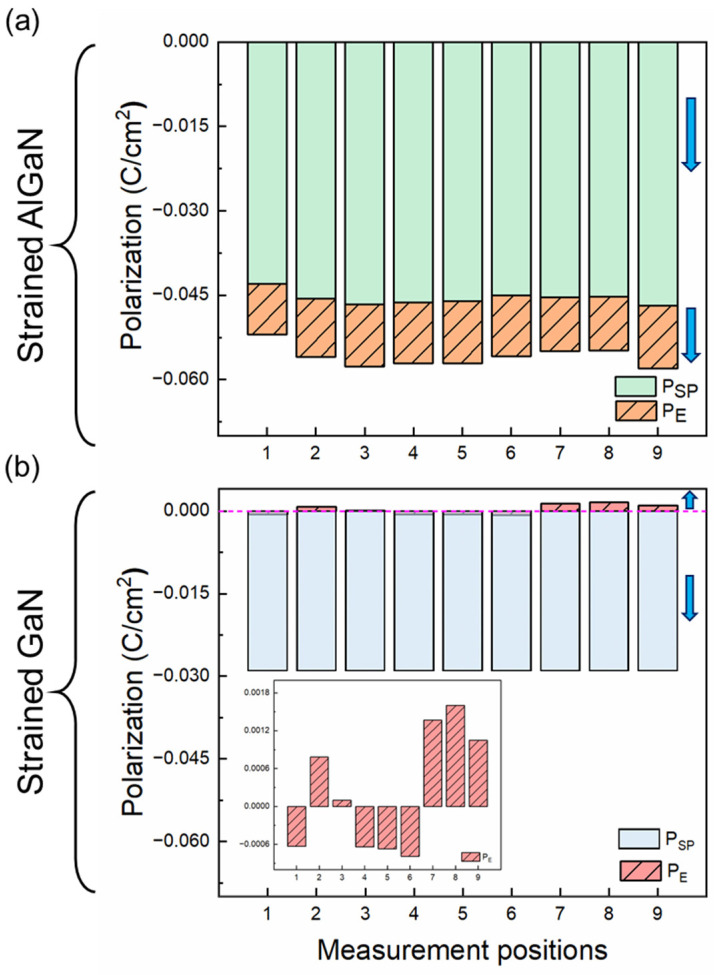
The polarization effect of AlGaN/GaN heterostructures at different measurement positions. (**a**) Spontaneous polarization (*P_SP_*) and piezoelectric polarization (*P_E_*) in strained AlGaN and (**b**) in strained GaN. The arrow represents the direction of polarization.

**Figure 8 micromachines-15-00536-f008:**
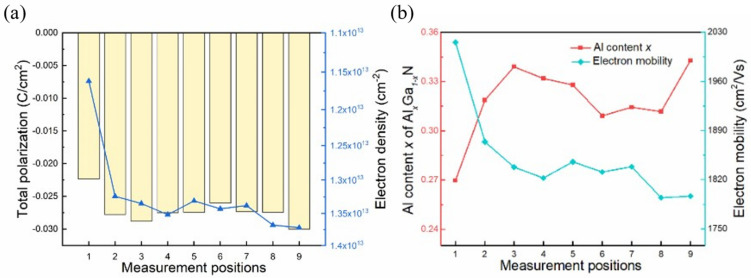
(**a**) The relationship between 2DEG density and fixed polarization charge density of AlGaN/GaN heterostructures at different measurement positions. (**b**) The relationship between electron mobility and the Al content *x* of the AlGaN barrier layer.

**Table 1 micromachines-15-00536-t001:** The FWHM of different locations in the AlGaN/GaN HEMT on Si(111).

Measurement Locations	(002) FWHM (Arcsec)	(102) FWHM (Arcsec)
1	505.91	809.93
2	515.48	795.78
3	516.17	794.99
4	516.02	798.84
5	516.60	795.24
6	513.14	788.98
7	509.94	786.17
8	514.80	786.60
9	514.40	790.31

**Table 2 micromachines-15-00536-t002:** Electrical properties of materials.

		GaN	AlN	Al*_x_*Ga_1−*x*_N
Piezoelectric coefficients (C/m^2^)	*e* _31_	−0.49	−0.6	−0.49 − 0.11*x*
	*e* _33_	0.73	1.46	0.73 + 0.73*x*
Elastic constants (Gpa)	*C* _11_	367	396	367 + 29*x*
	*C* _12_	135	137	135 + 2*x*
	*C* _13_	105	108	103 + 5*x*
	*C* _33_	405	373	405 − 32*x*
Spontaneous polarization (C/m^2^)	*P_SP_*	−0.029	−0.081	−0.052*x* − 0.029

## Data Availability

The datasets generated and supporting the findings of this article are obtainable from the corresponding author upon reasonable request.

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
