# Peer review of "The Effect of the Barrier Layer on the Uniformity of the Transport Characteristics of AlGaN/GaN Heterostructures on HR-Si(111)"

_micromachines, 2024, doi:10.3390/mi15040536_

Round 1

Reviewer 1 Report

Comments and Suggestions for Authors

This work investigated the transport properties of AlGaN/GaN heterostructure grown on HR-Si substrate. Please find my comments below.

1)    The reviewer noted that some parameters in the equations are undefined or unexplained. It is recommended that authors should carefully check all equations and clearly define all parameters.

2)    The authors focus on AlGaN/GaN HEMTs. The reviewer recommends adding additional electrical data, such as IV curves, to facilitate a better understanding of the differences and enhance the completeness of the manuscript.

3)    The manuscript primarily centers on AlGaN/GaN HEMTs. The reviewer recommends adding the following references to help readers understand their advantages. Doi: 10.3390/mi9120658; 10.1109/TNS.2023.3336836; 10.1109/TDMR.2020.2986401; 10.3390/electronics7120377.

4)    Please include a brief discussion on the reliability of AlGaN/GaN HEMTs as discussed in the manuscript.

5)    There are some typos present in the manuscript. The reviewer suggests that the authors should thoroughly review the entire manuscript and make the necessary corrections.

Comments on the Quality of English Language

Minor editing of English language required.

Reviewer 2 Report

Comments and Suggestions for Authors

The transport properties of AlGaN/GaN heterostructure grown on HR-Si substrate were investigated. It was proposed that Al content and stress in AlGaN barrier layer are the main factor affecting the uniformity of transport properties. This topic can be interesting for the community. May be some questions arise to the novelty because the made conclusions can be expected. Nevertheless, in my opinion it can be accepted for publication. The only remark is that in Fig. 2c and 8a the 2DEG density in the point 1 is the lowest one while in the text it is written that it is higher.

Reviewer 3 Report

Comments and Suggestions for Authors

In the process of industrializing large-scale Si-based AlGaN/GaN HEMTs, uniformity issues are a key challenge. Although the solution to this problem mainly depends on the continuous improvement of equipment and epitaxial processes, in-depth understanding and comprehensive analysis of uniformity, especially electrical performance uniformity, will help to address this issue more effectively. The research content of this paper may attract the attention of practitioners in the field of Si-based AlGaN/GaN HEMT research and development, providing valuable references for the further development of the industry.

1.        References to figures need to be standardized, and attention should be paid to singular and plural forms. For example, "Figure 5b1" should be written as "Figure 5(b1)", and "Figure 3a1-a3" should be written as "Figures 3(a1)-(a3)."

2.        In Figure 5, please specify which formulas were used to calculate the Al content (x) and in-plane stress.

3.        Both figures in Figure 6 require separate textual descriptions.

4.        The sentence "Figure 6 shows wafer bow of AlGaN/GaN HEMT on HR-Si substrate, measured by Flatness Analyzer. The bow value is -28.7μm." does not match the content displayed in Figure 6, as there is no -28.7μm value indicated. Please clarify this point.

5.        In Figure 8(a), there are numerical values on the right side without annotations. Please provide explanations for these values.

6.        The motivation of this work should be emphasized in the introduction part.

Comments on the Quality of English Language

Should be improved.

Round 2

Reviewer 1 Report

Comments and Suggestions for Authors

All my comments have been well addressed. Thanks!

Comments on the Quality of English Language

Minor editing of English language required